# Clinical development success rates and social value of pediatric Phase 1 trials in oncology

**Mateusz T. Wasylewski**[1], **Karolina Strzebonska**[1], **Magdalena Koperny**[1],
**Maciej Polak**[1,2], **Jonathan Kimmelman**[3], **Marcin Waligora**[1]*

**1** Dept. of Philosophy and Bioethics, REMEDY, Research Ethics in Medicine Study Group, Jagiellonian University Medical College, Krakow, Poland, **2** Dept. of Epidemiology and Population Studies, Jagiellonian University Medical College, Krakow, Poland, **3** Studies of Translation, Ethics and Medicine (STREAM), Biomedical Ethics Unit, McGill University, Montreal, Canada

* m.waligora@uj.edu.pl

**Data Availability Statement:** All relevant data are available from the Open Science Framework database (osf.io/9q6cw/).

## Abstract

### Objectives

Drug development trials must fulfill social value requirement but no estimates of value provided by pediatric Phase 1 trials in oncology exist. These trials involve a particularly vulnerable population. Our objective was to assess of surrogates of social value of Phase 1 trials performed in pediatric oncology: rates of approval of tested interventions, transition to further phases of testing and citation in subsequent primary research reports.

### Methods

We performed an analysis on a subset of eligible trials included in a previous meta-analysis. That study systematically searched EMBASE and PubMed for small sample size, non-randomized, dose escalation pediatric cancer Phase 1 studies of any malignancy, assessing chemotherapy and/or targeted therapy and looked at risk and benefit. The current analysis assessed all studies in that review published between January 1st 2004 and December 31st 2013 for predictors of social value. This time range allowed for at least five years of subsequent development activity. Sources of data included FDA and EMA medicine databases (for approval), ClinicalTrials.gov and EU Clinical Trials Register (for transition) and Google Scholar (for citation).

### Results

One hundred thirty-nine trials enrolling 3814 patients met the eligibility criteria. Seven trials (5%) led to drugs being registered for pediatric use in therapy of cancer. Fifty-two (37%) transitioned to later phases of pediatric oncology trials according to ClinicalTrials.gov and/or EU Register. Over 90% of trials were cited by at least one subsequent primary research report or systematic review. Most of the citations were preclinical studies.

**Funding:** This study was funded by the National Science Center, Poland, UMO-2015/18/E/HS1/00354 (www.ncn.gov.pl). Authors who received the funding: MTW, KS, MK, MP, MW (PI). The funders had no role in study design, data collection and analysis, decision to publish, or preparation of the manuscript.

**Competing interests:** Marcin Waligora reports personal fees from Advisory Bioethics Council, Sanofi outside the submitted work. This does not alter our adherence to PLOS ONE policies on sharing data and materials. Other authors have declared that no competing interests exist.

## Conclusions

Our analysis shows that treatments tested in pediatric Phase 1 trials in oncology have low rates of regulatory approval. However, a large proportion of Phase 1 trials inform further testing and development of tested interventions.

## Introduction

Phase 1 clinical trials in oncology are designed to test safety, identify dose recommended for further testing and probe the pharmacologic and pharmacodynamic performance of new treatments [1,2]. The probability that new cancer drugs tested in Phase 1 will reach regulatory approval is reported to be around 3–7% [3–5]. No estimates exist for oncology agents targeting pediatric patients during Phase 1 trials [6]. Children are a unique group of participants and are considered vulnerable. Pediatric trials are evaluated with enhanced caution and must adhere to stricter standards than their counterparts with adult participants [7]. Most regulations allow only minimal risk or require the prospect of direct benefit for participants [8–10].

At the same time all drug development trials must fulfill social value requirement [11]. For example, Council for International Organizations of Medical Sciences (CIOMS) guidelines requires social and scientific value for all studies. It states that studies have to be "scientifically sound, build on an adequate prior knowledge base and (. . .) likely to generate valuable information" [9]. Since (beside identifying toxicities) the goal of Phase 1 studies is to develop clinically useful treatments, one key proxy for measuring social value is the extent to which treatments tested in Phase 1 trials advance to later phases and/or regulatory approval. A recent meta-analysis of Phase 1 trials in pediatric oncology showed that they provide limited direct benefit for participants [12,13]. On average every participant experienced at least one serious adverse event that is associated with treatment, and 1 in 50 died from such an event [12]. Therefore the justification of risk rests heavily on demonstrating social value.

The social value of pediatric Phase 1 trials has not, to our knowledge, been subject to systematic analysis. In what follows we assess clinical development success rates and other proxies of social value for a sample of pediatric Phase 1 trials in oncology to examine how frequently such trials influence clinical development.

## Methods

Our study protocol was prospectively registered in PROSPERO database (CRD42018106213) [14].

### Study selection and eligibility criteria

This analysis was based on a subset of trials analyzed in our previous systematic review of risk and benefit in Phase 1 pediatric clinical trials [12], (Fig 1). In short, the previous study involved a systematic search of PubMed and EMBASE to capture all Phase 1 pediatric trials in oncology of any malignancy, assessing chemotherapy and/or targeted therapy published between January 1st 2004 and March 1st 2015. We excluded trials testing radiotherapies, surgeries or cell therapies. We defined Phase 1 as a small sample size, non-randomized, dose escalation study that defined the recommended dose for subsequent study of a new drug in each schedule tested [12]. Included trials were considered "pediatric" if all or most participants were below

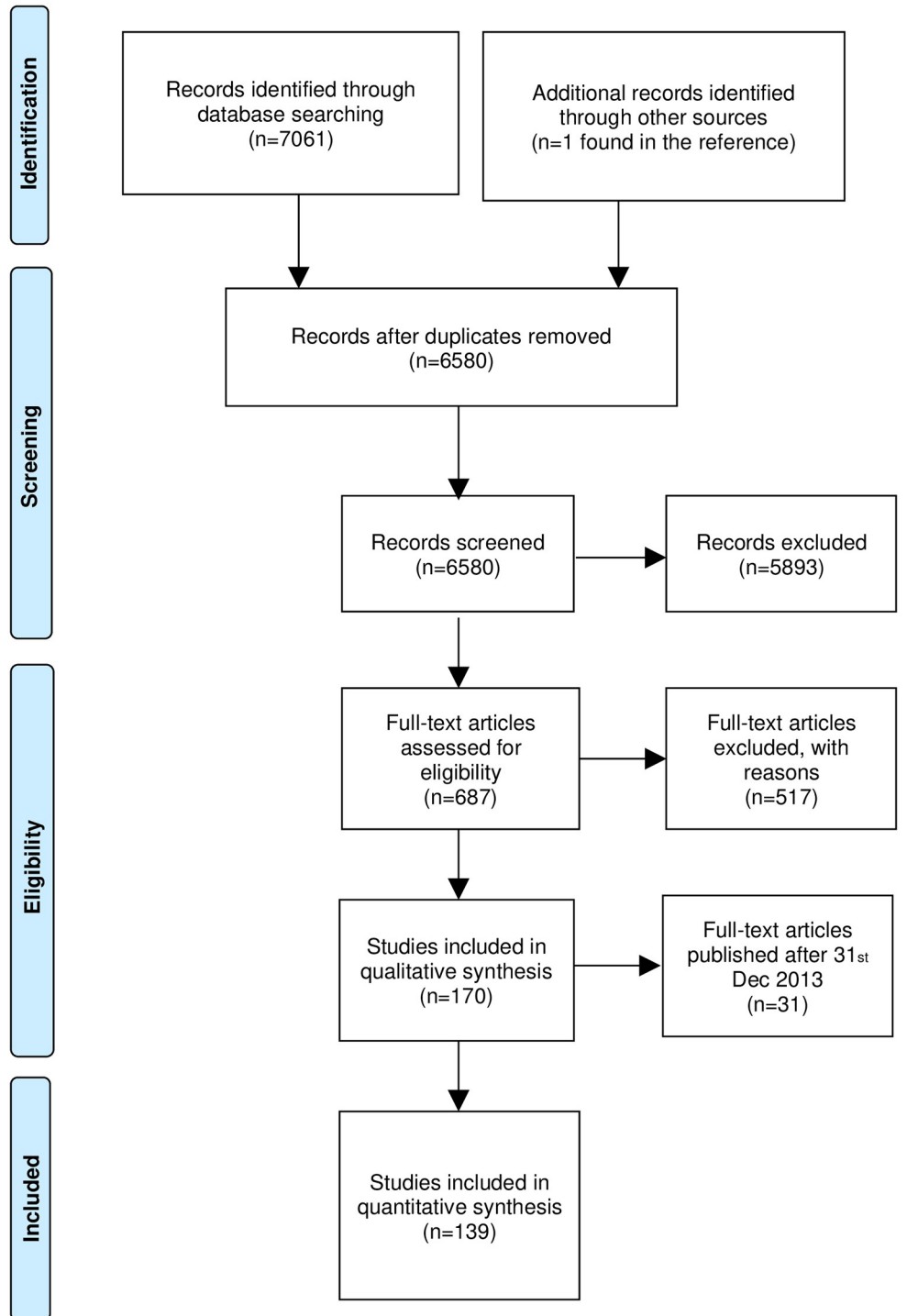

**Fig 1. PRISMA flowchart detailing stages of the review process.** Number of records, full-text articles and studies included at each stage of the review process.

21 years of age. Choosing 21 years as an age threshold enabled us to encompass both EMA and FDA definitions of pediatric participants.

To ensure that Phase 1 trials included in the present study had an adequate time-frame for graduation to later phases of testing or to be cited in literature since closure, for further

analysis we selected those studies from our previous sample for which we had at least five years of follow up since publication. This involved including in the current analysis a total of 139 trials form the previous dataset that were published between January 1$^{st}$ 2004 and December 31$^{st}$ 2013. Note that the use of publication date for defining our cohort, as opposed to trial close date, was driven by the absence of a report of the latter in many publications. The PRISMA chart shows the flow of studies (Fig 1).

## Assessment of approval rate

Our first measure of impact was approval success, which we defined as registration of the tested treatment by FDA or/and EMA for pediatric oncologic indications either as an approval or a revision of label. Registration status was determined by searching FDA [15] and EMA [16] medicine databases for drugs tested in our sample of trials. The regulatory documents for drugs approved by either agency were then reviewed to identify the target population (pediatric or non-pediatric), indication (oncologic or non-oncologic) and date of the first approval. Drugs were classified as approved for pediatric use if there was a clear reference to pediatric population in "indications" or "dosing" sections of regulatory documents. If a Phase 1 study tested more than one drug, approval status for all drugs was determined. The trial regimen was then classified as "successful" if the whole combination (all of the investigated drugs) were approved by FDA and/or EMA after study publication. Registration documents often did not identify specific Phase 1 trials leading to an approval. Therefore the matching of a Phase 1 trial to an approval was made by two independent coders (KS, MTW) and verified by a third coder (MK).

## Assessment of transition rate

Our second measure of impact was transition, which was defined as advancing to later phases of clinical trials. We searched the ClinicalTrials.gov and the EU Clinical Trials databases to find further pediatric studies of drugs tested in the Phase 1 trials. We determined that Phase 1 study regimen had advanced to later phases if we were able to find at least one Phase 2, 3 or 4 trial that tested exactly the same drugs or drug combinations (of all investigated drugs) in at least one oncological pediatric indication tested in the Phase 1 trial. To qualify, trials must have started after the original Phase 1 study publication. We supplemented the search of ClinicalTrials.gov with searching Google Scholar for Phase 2, 3 or 4 pediatric trials that cited Phase 1 trials from our group and tested the same drug/s.

## Assessment of the citation patterns

Our third measure of impact was influence on subsequent research. This was measured by citation of Phase 1 trial publications in subsequent primary research reports. We used Google Scholar to identify all citations of our sample of Phase 1 trials. Google Scholar was used to maximize capture of citations (including e-pubs, publications ahead of print, conference abstracts and the gray literature) which are not necessarily covered by other databases [17]. We screened full texts of publications that cited original Phase 1 trials included in our group. Searches were not limited by language. Citing articles were then assigned to categories representing different types of social value: 1) further phase trials of the same drug or combination, clinical practice guidelines; 2) other pediatric or adult Phase 1 trials using the same drugs or combinations as original Phase 1 trials; 3) preclinical studies; 4) clinical trials for other drugs or other non-trial biomedical research (i.e. pediatric and adult case studies and patient cohorts, non-interventional studies, retro-/prospective studies, subgroup analyses, surveillance studies, modelling, diagnostic methodology studies, biomarker studies, studies on tissue samples taken from

treated patients or pharmacokinetics in treated patients and in response to treatment); 5) systematic reviews. We excluded citations other than primary reports, i.e.: non-systematic literature reviews, editorials, letters, commentaries, book chapters, various patent or regulatory documents not related directly to trials etc.

## Data extraction

In our previous systematic review of risk and benefit of Phase 1 trials in pediatric oncology we created a database of information and outcomes extracted from each study. We retrieved from this database data on study design, funding, intervention, outcomes and the timing of pediatric testing relative to adult testing for every trial included in the current analysis. For the outcomes of approval rate, transition rate and citation patterns we created and piloted a new extraction form. Data from online FDA, EMA and ClinicalTrials.gov and EU Clinical Trials Register databases were extracted by one experienced reviewer (KS or MTW) and checked by another experienced reviewer (MK). Disagreements were resolved by discussion, and when necessary a third person—an arbiter—was involved (MW). Data on citation patterns were extracted by 1 reviewer (MTW). Screening of the trials and/or approval documents was performed by one experienced reviewer (KS or MTW) and checked by another experienced reviewer (MK). We extracted data on approval status and citations [12].

## Statistical analysis

Data on study characteristics were summarized using absolute frequencies. Outcomes for a) approval for pediatric population by FDA or/and EMA, b) study being cited in other publications and c) advancement to further phases of clinical trials were compared by Pearson's chi-squared test or by Fisher's exact test (if 20% of cells have expected count less than 5) in subgroups of studies according to their characteristics. The calculations and testing were performed using IBM SPSS Statistics for Windows, version 25 (IBM Corp., Armonk, N.Y., USA). We hypothesized that the number of drugs tested and previous approval status may impact the probability of approval and transitional success. We prespecified 'significance' as $p < 0.05$. However, all inferential testing was exploratory only and we did not adjust for multiple hypothesis testing.

## Results

### Characteristics of the trial group

There were 170 trials included in our previous review. We determined the registration status of each drug tested in those studies by searching Food and Drug Administration (FDA) and European Medicines Agency (EMA) registers of approved medicines. We found that 6 drugs tested in 7 trials were approved by at least one agency for pediatric use in oncologic conditions (S1 Table). We also determined that all registrations had been obtained within five years since trial publication. From our previous systematic review we retrieved a sample of 139 Phase 1 pediatric oncology trials (enrolling 3 814 patients) that allowed for a 5-year follow-up of transition and citation outcomes from publication date. These were included in the final analysis presented here.

Characteristics of included studies are shown in Table 1. Most trials were described as Phase 1 only (94%), enrolled patients with solid tumors (78%) and tested only one drug (60%). Most studies (101 of 139; 73%) in our sample were initiated after Phase 1 trials of the same substance/s had already been completed in adult patients. Most studies (81%) in our sample recommended proceeding to a later trial; 6 studies (4%) recommended against further testing

**Table 1. Characteristics of included studies.**

| | | Number of studies (%) |
|---|---|---|
| **No. studies** | | 139 (100) |
| **Study definition** | **Phase 1** | 130 (94) |
| | **Phase 1/2** | 9 (6) |
| **Type of tumor** | **Solid** | 108 (78) |
| | **Hematological** | 23 (17) |
| | **Both** | 8 (6) |
| **Initiated after study in adults?** | **Ongoing study in adults** | 1 (1) |
| | **After adults Phase 1** | 101 (73) |
| | **Unclear** | 11 (8) |
| | **Not reported** | 24 (17) |
| | **No study in adults** | 2 (1) |
| **Was Phase 2 recommended?** | **Yes** | 112 (81) |
| | **No** | 6 (4) |
| | **Unclear or not reported** | 21 (15) |
| **Number of drugs** | **1 drug** | 84 (60) |
| | **2 or more drugs** | 55 (40) |
| **Funding** | **Private for profit** | 7 (5) |
| | **Private not for profit** | 1 (1) |
| | **Public** | 39 (28) |
| | **Mixed** | 64 (46) |
| | **Not funded** | 1 (1) |
| | **Not reported** | 27 (19) |

Percentages show proportion of type of trial (e.g., Phase 1) in each category (e.g., study definition)

(Table 1). S2 Table presents a list of drugs and drug combinations tested in the trials included in our study.

## Approval rate

Our primary outcome was the approval rate (Table 2). Seven out of 139 trials (5%) had their investigational drugs registered for pediatric use in therapy of cancer; characteristics of those trials are shown in S1 Table. There was a significant relationship between type of malignancy studied in the trial (solid or hematological) and the subsequent regulatory approval; 19% of hematological drugs were ultimately approved vs 3% of solid tumor drugs (p = 0.025). Every trial that led to approval was a single drug study. There was no relationship between a drug's previous approval by FDA or EMA for use in adult population and probability of approval for children (4% vs. 8% for non-adult approved, p = 0.26, Table 2).

## Transition rate

Fifty-two out of 139 trials (37%) transitioned to later phases of pediatric oncology trials according to one or both databases (Table 3). The separate results for each database are summarized in the S4 Table. Eighteen regimens were found both on ClinicalTrials.gov and through EU Clinical Trials Register. Fifteen trials were identified in both registers at once as having transitioned to later phases of testing. Through Google Scholar we determined that 27 of Phase 1 trials included in our analysis were cited by Phase 2 or 3 primary research reports of the same drugs or drug combinations. The proportion of treatments transitioning according to

**Table 2. Rate of approval for pediatric population by FDA or/and EMA.**

| Approved for pediatric population by FDA or/and EMA | | Yes | No | p-value |
|---|---|---|---|---|
| | | Number of studies (%)* | | |
| | TOTAL | 7 (5) | 128 (95) | |
| Type of tumor | Solid | 3 (3) | 103 (97) | 0.03 |
| | Hematological | 4 (19) | 17 (81) | |
| | Both | 0 (0) | 8 (100) | |
| Number of drugs | 1 drug | 7 (9) | 74 (91) | 0.04 |
| | 2 or more drugs | 0 (0) | 54 (100) | |
| Drug/s generally approved by FDA or EMA before study publication | Yes | 3 (4) | 82 (96) | 0.26 |
| | No | 4 (8) | 46 (92) | |

* Four trials were excluded from the analysis due to not being applicable or an unclear status

p-values were calculated using Fisher's exact test

ClinicalTrials.gov was not significantly correlated with type of tumor, number of drugs studied or being previously approved for adults by FDA or EMA. The proportion of treatments transitioning according to EU Register was not significantly correlated with type of tumor or being previously approved for adults by FDA or EMA but was correlated with the number of drugs studied (19% single-drug studies vs. 3% multiple-drug studies; $p = 0.008$). Transition was significantly correlated with type of tumor (15% solid tumor studies vs. 26% hematological studies; $p = 0.04$) and being previously approved for adults by FDA or EMA (25% approved vs. 10% not previously approved; $p = 0.04$). Overall, the transition to further phases was significantly correlated with the number of drugs studied (46% single-drug studies vs. 23% multiple-drug studies; $p = 0.007$) and being previously approved for adults by FDA or EMA (45% approved vs. 21% not previously approved; $p = 0.01$).

## Citation patterns

Almost every reviewed trial (94%) was cited by at least one subsequent primary research report or systematic review. Type of malignancy treated in the trial or previous approval for adult populations did not have statistically significant impact on whether a study was cited. Trials that studied only one drug were more likely to be cited than trials with two or more drugs

**Table 3. Phase 1 transition success, defined as the number of treatments that were advanced to the next pediatric phases.**

| Advanced to the next pediatric phases | | Yes | No | p-value |
|---|---|---|---|---|
| | | Number of studies (%) | | |
| | TOTAL | 52 (37) | 87 (63) | |
| Type of tumor | Solid | 37 (34) | 71 (66) | 0.08* |
| | Hematological | 9 (39) | 14 (61) | |
| | Both | 6 (75) | 2 (25) | |
| Number of drugs | 1 drug | 39 (46) | 45 (54) | 0.007** |
| | 2 or more drugs | 13 (24) | 42 (76) | |
| Drug/s generally approved by FDA or EMA before study publication | Yes | 40 (45) | 49 (55) | 0.01** |
| | No | 12 (24) | 38 (76) | |

* p value for differences between type of tumor categories provided by Fisher's exact test

** p value for differences between number of drugs and previous approval categories provided by Chi-squared test

percentages show proportion of outcome in each sub-category (e.g. solid tumor)

tested at once (98% vs 87%, Fisher's exact test p-value = 0.015). These results are summarized in S5 Table.

The S6 Table shows citation patterns of Phase 1 trials included in our study. Most of citations were trials for other drugs or non-trial biomedical research (45%). Other prevalent citations were by preclinical studies (41%), systematic reviews (9%), other Phase 1 using the same drugs (1%) as well as Phase 2 or 3 trials of the same drugs clinical practice guidelines (4%).

## Discussion

The overall percentage of Phase 1 pediatric oncology trials included in our study that resulted in a regulatory approval was approximately 5%. This estimate is similar to the global success rate for oncology drug development previously reported from first-in-human trials to approval (5%-10%) [3,18]. More trials assessing treatment for hematological malignancies led to approval than studies in solid tumors. This finding was similar to the one reported in another study [6], and may relate to higher rate of objective response observed in hematological trials (28%) when compared to solid tumor trials (3%) as shown by our previous systematic review [12]. Since every trial that led to an approval was a single-drug study our findings suggest that studies testing only one drug in a trial have a greater impact on care than trials testing multiple drugs. This finding can be related with an ongoing increase in development of targeted treatments in oncology, which are predominantly single-drug therapies [12]. We classified trial regimens as approved only if every drug investigated was registered as approved by at least one agency (either before or after the trial completion). While this made single-drug trials more likely to lead to successful approval, it was done to ensure that the exact treatment tested in the trial is available to be used in a clinical setting (i.e. every drug has obtained registration and approval). These findings are similar to the results of a recent study which examined the impact of various variables for their influence on Phase 2 pediatric oncology clinical trials [19]. The authors found that Phase 2 trials initiated based on the results of adult trials were less likely to produce positive results. We however did not find any significant relationship between previous approval for adult population and probability of subsequent drug approval for children in our analysis. Previous adult approval was not significantly associated with a higher probability of an intervention receiving a pediatric approval (Table 2). The current practice of delaying pediatric trials until after adult trials have been concluded is based in part on the reasoning that already available adult data will facilitate pediatric trials that are safer and more beneficial thereby leading to more drug approvals [20]. Our data do not support this view. It is however possible for a multiple-drug trial in our analysis to be primarily testing one drug on top of a backbone of standard treatment. Given promising results, only the drug of interest could then transition to further testing—such situation could impact our estimation of transitional success.

One of the aims of Phase 1 studies is to ultimately lead to development of clinically useful treatments and therefore provide social value. There are several ways of understanding social value in clinical research [9,11,21–23]. The most common definition is based on the idea that drug development is a sequential pipeline of research. The CIOMS guidelines recognize both social and scientific value of clinical trials (the importance of the information produced and the ability to produce reliable, valid information respectively). According to CIOMS guidelines trials have to be "scientifically sound, build on an adequate prior knowledge base and are likely to generate valuable information" [9]. In this model a study is valuable when it facilitates next stages of research and provides enough evidence to change clinical practice. In our study we used transitional success (phase transition) as proxy of scientific value of a study. The scientific impact of the trial is also reflected in its citations in subsequent publications. Almost a half of

trials in our sample had their treatments advanced to the next pediatric phase and almost all trials in our sample were cited by other publications. Among 2 060 individual citations of 139 studies included in this analysis, we found 14 clinical practice guidelines (0.68%), 69 (3.4%) Phase 2 trials and and 3 (0.15%) Phase 3 studies. We decided to lump them together into one category as a surrogate of influence on clinical practice. Pediatric Phase 1 trials in oncology may influence drug development in various ways other than leading to regulatory approval. They can inform decisions to modify the current phase of research, prompt further preclinical research or motivate research in other scientific areas. The linear model of drug development and value does not reflect these influences. For example Crizotinib, one of the drugs tested in our sample, did not fulfill our criteria of approval success (it was not approved for children for an oncologic indication within 5 years since study publication) but its citations indicate that data from this early trial strongly influenced pre-clinical research (S2 Table). The article was cited by over 70 pre-clinical studies and by 27 publications of case reports and cohort studies of pediatric cancer patients.

Our analysis has several limitations. First, our group of 139 trials involved 5 years of follow-up from publication of the Phase 1 trial. On the one hand, changes in transition or approval rates occurring in the last five years will not be reflected in our estimates. On the other hand, any phase transition or influence on subsequent publications occurring outside that time-frame would not be reflected in our analysis. After our search of approval status we found that a gap of 5 years since publication was sufficient to accommodate for the lag between publication of Phase 1 results and approval for pediatric indications. We therefore decided this gap was also sufficient to accommodate for phase advancement or citation. It is possible, however, that some regulatory approvals or phase transitions might have occurred after a Phase 1 trial was complete but before it was published. As such, our estimates might have undercounted some approvals and transitions. Second, our analyses do not include all types of trials: we excluded antiviral agents, non-specific immunotherapies, cell therapies and other agents when tested together with surgery or radiotherapy. Third, we included only published Phase 1 trials. However, it is estimated that about a third of cancer Phase 1 trials may go unpublished [24,25]. We suspect this may result in estimates that were biased towards overestimating approval success and graduation proportions. Third, our study focused only on labeling information provided by two major medicine agencies and regulators. We understand however that such information (e.g. approval dates) may sometimes be incorrect [26] and they do not consider off-label use. This may potentially underestimate the influence and social value of assessed trials, as many front line treatments that are standard of care for childhood cancers are used off-label. While there are economic reasons for manufacturers of these drugs to forgo pursuit of approval in specific pediatric indications, there are at the same time FDA and EMA policies that encourage pharmaceutical companies to seek approval in pediatric indications [27,28]. While this approach is a limitation of this analysis, it is very difficult to collect reliable data on off-label use—especially on a larger scale, as attempted here. For example, using other benchmarks such as treatment guidelines and recommendations would be difficult, given the paucity of specifically pediatric guidelines found by our analysis. Additionally, while searching for evidence of Phase 1 trials transition to further pediatric phases we may have lost some trials that were started between Phase 1 completion and publication. The EU Clinical Trials Register contains information on trials started after May 1st 2004, while our study analyzed trials published between January 1st 2004. In our sample there were 6 trials published between January and May 2004, 4 of those we already identified through ClinicalTrials.gov as having transitioned. On the other hand, the register also provides information about older pediatric trials covered by an EU marketing authorization. It is worth noting that regulatory approval and phase transition are not the only metrics of success. In our view they allow however to assess global rates

of approval and transitional success of Phase 1 trials. The results of our analysis of phase transition may also be influenced by factors other than drug efficacy and toxicity. The development of the drug may be stopped by the sponsor due to economic constraints or a subsequent study may be unable to recruit sufficient number of participants. The fact that some recent cancer drugs have been directly developed using Phase 1/2, seamless trial approaches may be another limitation of our approach to estimating transition success [29], though we are not aware of any pediatric drug approvals in this period that resulted from a seamless approach.

## Conclusions

Our analysis shows that approximately 1 in 20 Phase 1 pediatric trials leads to regulatory approval. However, our analysis also suggests there is a large proportion of Phase 1 trials inform further development of tested interventions. Almost half of the treatments tested in Phase 1 transitioned to further phases of clinical trials. Even if they did not ultimately lead to regulatory approval the trials in question facilitate continuous iterations and further testing that are very important for developments in biomedical research. The trials in our sample were also widely cited in primary research reports. This analysis may be helpful for understanding the frequency and magnitude of social and scientific value associated with pediatric Phase 1 trials. It may also provide a basis for communicating to prospective research participants the prospects that their trial participation will lead to advances in treatment.

## Supporting information

**S1 Table. Trials that achieved approval.**
(DOCX)

**S2 Table. Drugs and drug combinations used in included studies, approval status.**
(DOCX)

**S3 Table. Transitions status of trials included in the review.**
(DOCX)

**S4 Table. Phase 1 transition success, defined as the number of treatments that were advanced to the next pediatric phases according to ClinicalTrials.gov, EU Clinical Trials Register and Google Scholar.**
(DOCX)

**S5 Table. Number of trials that were cited by primary research reports and systematic reviews.**
(DOCX)

**S6 Table. Citation patterns of Phase 1 trials.**
(DOCX)

**S1 Text.**
(TXT)

## Author Contributions

**Conceptualization:** Mateusz T. Wasylewski, Jonathan Kimmelman, Marcin Waligora.

**Data curation:** Mateusz T. Wasylewski, Karolina Strzebonska, Magdalena Koperny.

**Formal analysis:** Maciej Polak.

**Funding acquisition:** Marcin Waligora.

**Investigation:** Mateusz T. Wasylewski, Karolina Strzebonska, Magdalena Koperny.

**Methodology:** Mateusz T. Wasylewski, Karolina Strzebonska, Maciej Polak, Jonathan Kimmelman, Marcin Waligora.

**Project administration:** Mateusz T. Wasylewski, Marcin Waligora.

**Software:** Maciej Polak.

**Supervision:** Jonathan Kimmelman, Marcin Waligora.

**Validation:** Mateusz T. Wasylewski, Karolina Strzebonska, Marcin Waligora.

**Visualization:** Mateusz T. Wasylewski, Karolina Strzebonska.

**Writing – original draft:** Mateusz T. Wasylewski.

**Writing – review & editing:** Mateusz T. Wasylewski, Karolina Strzebonska, Magdalena Koperny, Maciej Polak, Jonathan Kimmelman, Marcin Waligora.

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
