## [Decision Letter · Decision Letter 0]

10 Mar 2020

PONE-D-19-32233

Clinical development success rates and social value of pediatric Phase 1 trials in oncology: A meta‑analysis

PLOS ONE

Dear Dr Waligora,

Thank you for submitting your manuscript to PLOS ONE. After careful consideration, we feel that it has merit but does not fully meet PLOS ONE’s publication criteria as it currently stands. Therefore, we invite you to submit a revised version of the manuscript that addresses the points raised during the review process.

We would appreciate receiving your revised manuscript by Apr 24 2020 11:59PM. To enhance the reproducibility of your results, we recommend that if applicable you deposit your laboratory protocols in protocols.io, where a protocol can be assigned its own identifier (DOI) such that it can be cited independently in the future. For instructions see: http://journals.plos.org/plosone/s/submission-guidelines#loc-laboratory-protocols

We look forward to receiving your revised manuscript.

Kind regards,

Spyridon N. Papageorgiou, DDS, Dr Med Dent

Academic Editor

PLOS ONE

Journal Requirements:

"I have read the journal's policy and the authors of this manuscript have the following competing interests: Marcin Waligora reports personal fees from Advisory Bioethics Council, Sanofi outside the submitted work. Other authors have declared that no competing interests exist."

Reviewers' comments:

Reviewer's Responses to Questions

**Comments to the Author**

1. Is the manuscript technically sound, and do the data support the conclusions?

Reviewer #1: Yes

Reviewer #2: Partly

Reviewer #3: Yes

Reviewer #4: Yes

2. Has the statistical analysis been performed appropriately and rigorously? 

Reviewer #1: Yes

Reviewer #2: No

Reviewer #3: No

Reviewer #4: Yes

3. Have the authors made all data underlying the findings in their manuscript fully available?

Reviewer #1: Yes

Reviewer #2: Yes

Reviewer #3: Yes

Reviewer #4: No

4. Is the manuscript presented in an intelligible fashion and written in standard English?

Reviewer #1: Yes

Reviewer #2: No

Reviewer #3: Yes

Reviewer #4: Yes

5. Review Comments to the Author

Reviewer #1: Waligora and colleagues submitted an investigation of the social and scientific value of pediatric phase I studies based on a previous systematic review of the group. The manuscript is well written and clear. The provided information is of value to the scientific community. I only have some minor comments that hopefully help to improve the manuscript

1. In your section statistical analysis, you mention to use techniques of meta-analysis to investigate the impact of previous EMA or FDA approval, but I cannot find that in your results section. Please clarify, because this would also impact on your manuscript title. Where is the meta-analysis?

2. Have you considered conducting a time to event analysis? E.g. time from publication of the phase I study to citation? By this you could additionally and easily visualize your findings.

3. I suggest to provide more information on the type of drugs; not only for those that have been approved by FDA/EMA later on, but for all that have been tested in your included phase I trials.

4. “Being previously approved for adults by FDA or EMA had no significant effect on progression to later phases of trials (p=.657 and p=.28 for ClinicalTrials.gov and EU Clinical 207 Trials Register respectively).” Please also report the frequencies with percentages, not only the p-values. You do that in the tables, but would be great to also see that in the text.

Reviewer #2: This paper included a sample of 139 pediatric phase 1 trials in oncology, previously found in a systematic review that retrieved studies from 2004 to 2013 (Waligora et al., 2018), in order to assess:

i. Approval rate (FDA or EMA) - named by the author as translational success,

ii. Rate of advance to further phases (phase 2, 3 or 4) - named by the author as transition success and,

iii. Citation patterns.

The authors found that 5% of pediatric phase 1 trials in oncology turned to approval in the FDA or EMA, despite a large proportion which have been tested in phase 2, 3 or 4. The trials were widely cited in primary research suggesting a scientific and social impact.

This is a very interesting research topic because pediatric population is vulnerable and few trials have been carried out in this population. Lack of published data increase the off-label use and evidence-based oriented practices is a challenge. United States of America and European Union are the head of a movement to increase pediatric research with special fundings and programs.

The paper aim is very important and open a discussion to understand the barriers to pediatric drug development.

I suggest improvements in the text and the methodology in order to clarify the reader and increase transparency:

Title - In my opinion, this paper did a statistical analysis with the data available, not a meta-analysis. A meta-analysis is defined as “a statistical procedure that integrates the results of several independent studies considered to be “combinable” (Egger M, Smith GD, Phillips AN, 1997). I suggest the title review;

Abstract/Objective - It would be more informative for the reader the definition of ‘translation’ or changing the term ‘translation’ for “approval rate in FDA or EMA”;

Abstract/Methods - Clarify for the reader that the cohort of 139 studies was systematically searched in a previous study (Waligora et al., 2018), in my opinion, it’s important to mention in the Abstract only the methodology of the current paper.

It is unclear to the reader this paragraph “We included small sample size, non‑randomized, dose escalation pediatric cancer phase 1 studies of any malignancy, that assessed chemotherapy and/or targeted therapy and were published between January 1st 2004 and December 31st 2013...”. Is this the inclusion criteria from the previous study? How was it related to other databases searched, FDA and EMA medicine databases, ClinicalTrials.gov and EU Clinical Trials Register or Google Scholar?

If this is the inclusion criteria of the previous systematic review, it would be more clear to maintain it in the beginning of the methodology.

Small sample isn’t an inclusion criteria in the paper methods.

Methods

I suggest that the authors clarify what is part of the previous study and what are the methods of the current one. Thus, I suggest that all the information regarding the systematic review, Waligora et al., 2018, remain in the session ‘Study selection and eligibility criteria’ (Pg4/Line73).

Moreover, a critical aspect of this study is the usage of 3 characteristics (‘Type of tumor’, ‘Number of drugs’ and ‘Drug/s generally approved by FDA or EMA before study publication’) in the results and for statistical analysis. However these topics were not mentioned in the methods. The authors should explain the rationale for this 3 characteristics inclusion and its impact in the aim and results of the study.

-Pg4/Line73 - It would be important to mention the databases searched in the previous systematic review (The authors mentioned it in the Abstract).

-Pg4/Line76 - I suggest a Phase 1 definition as their prior paper “Phase I studies, defined as “small sample size, non-randomized, dose escalation studies that defined the recommended dose for subsequent study of a new drug in each schedule tested”” (Waligora et al., 2018)

-Pg4/Line80 - It would be important to reference the pediatric age range, because it can diverge, e.g. the FDA consider pediatrics individuals under 18 years old and the American Academy of Pediatric define pediatrics individuals under 21 years old.

- Pg5/Line96 and Pg5/Line98-104 - Information about previous study and criteria for selection, I suggest to move to ‘Study selection and eligibility criteria’ (Pg4/Line73). How many studies were assessed, 170 or 139? In my understanding all outcomes should use the same timeframe previously specified from 2004 to 2013. The PRISMA flow in the Supplemment material made it clear however, in the text it would be important to fill this gap to the reader understanding.

- Pg5/Line96-98 “After determining the registration….(Table 1) - Table 1 and this paragraph are results. Describe the methods which information the authors will extract from the papers.

-Pg9/153 - I suggest an independent topic about ‘Data extraction’

-Pg9/158-9 - The authors mentioned data extraction related to: “study design, funding, intervention, outcomes and the timing of pediatric testing relative to adult testing as well as data on response and toxicity”. However only funding was reported as result. Please check data extraction.

In the results, it was collected data and performed statistical analysis for ‘Type of tumor’, ‘Number of drugs’ and ‘Drug/s generally approved by FDA or EMA before study publication’ for each outcome (i. Approval rate, ii. Posterior phases rate and iii.Citation pattern), despite it, none of these information were mentioned in the method.

In the PRISMA figure at the Supplemental material and reading Waligora et al., 2018, it’s clear that 139 studies is from 2004 to 2013 and 170 the number retrieved in the previous study from 2004 to 2013.

Results

-Pg10/173 - Clarify that ‘One hundred and thirty-nine Phase 1 pediatric oncology trials’ was retrieved from the past study.

- Pg12/209 - In Table 4, I suggest to merge information about Clinicaltrials.gov and EU Clinical Trials Registered. I understood that the analysis aims to access transition to further phases and not compare the protocol-registry databases. Moreover, it’s mandatory the previous registration in the database linked to the trial location, so ClinicalTrials.gov would be mandatory to studies carried out in the USA and EU CT in Europe. Thus, it would be more useful to show the data in a single column, Transition success (advance to further phases in Clinicaltrials.gov OR EU Clinical Trials Registered.)

Discussion

Pg12/227 - The authors might describe Phase 1 trials in details, ‘pediatric Phase 1 trials in oncology’.

Pg13/234 - It would be interesting if the author related this increase in single drugs approval with increase in target treatment in oncology that are in the great majority, a single drug. Multiple drugs approval are associated with conventional chemotherapy cytotoxic drugs, which have very few approvals in the past years.

Pg13/242 - This finding isn’t clear in the results.

The discussion is interesting and I additionally suggest that the authors include more discussion about the link between their results and the opportunities and challenges for pediatric research development.

Reviewer #3: The authors present the results of a meta-analysis of Phase I pediatric oncology clinical trials. This current study is based on a previous systematic review of risk and benefit of pediatric Phase I clinical trials. From that study, the authors have selected 139 studies for further analysis to assess surrogates of social value of Phase I trials in pediatric cancer patients.

The idea is interesting and the effort from the authors is laudable.

I have some comments for the authors.

INTRODUCTION:

LINE 46: When the authors state that there is not for oncology agents targeting pediatric patients, they may want to consider adding this reference: Neel D, Timing of first-in-child trials of FDA-approved oncology drugs. Eur J Cancer 2019.

METHODS:

Beyond the 139 selected studies I am missing a few Phase I trials that may fulfill the eligibility criteria but are not included within this short list (Table S3). The following are only some examples:

- Caruso DA, Orme LM, Neale AM, et al. Results of a phase 1 study utilizing monocyte-derived dendritic cells pulsed with tumor RNA in children and young adults with brain cancer. Neuro Oncol. 2004;6:236-246.

- Kramer K, Humm JL, Souweidane MM, et al. Phase I study of targeted radioimmunotherapy for leptomeningeal cancers using intra-Ommaya 131-I-3F8. J Clin Oncol. 2007;25:5465-5470.

- Gilman AL, Jacobsen C, Bunin N, et al. Phase I study of tandem high-dose chemotherapy with autologous peripheral blood stem cell rescue for children with recurrent brain tumors: a Pediatric Blood and MarrowTransplant Consortium study. Pediatr Blood Cancer. 2011;57:506-513.

In Table 1, the authors present a list of drugs that are approved for pediatric indications. In this table, the authors need to:

-In the column type of malignancy depict only the disease for which the drug is approved in children. For instance: Tablets for oral suspension of everolimus was granted by the FDA in 2012 for the treatment of pediatric and adult patients with tuberous sclerosis complex (TSC) who have subependymal giant cell astrocytoma (SEGA) that requires therapeutic intervention but cannot be curatively resected, but not the rest of the diseases that the authors describe in this column.

-For each drug, to add a reference in the table supporting the information about the specific approval indication

In this same table I have some comments:

- For the imatinib indication in pediatric brain tumors, can kindly the authors confirm the source of this information? I have not been able to find that imatinib is approved for such a number of indications as stated in this table. For double checking I have used this source: https://www.fda.gov/drugs/resources-information-approved-drugs/hematologyoncology-cancer-approvals-safety-notifications

- For the everolimus indication, the phase I from Fouladi did not include patients with SEGA, which is the final indication for this drug. This study, nevertheless, could contribute to the approval in 2012 of the everolimus tables for oral formulation, but in reality, this drug was already approved in 2010 for patients 3 years and older based on the pediatric and adult phase I/II trial (Krueger et al, NEJM 2010, NCT00411619). Furthermore, the approval in 2012 that the authors mention, is not based on the Phase I trial from Fouladi but rather in the Phase III randomized trial including children and adults (Franz, Lancet 2013, NCT00789828). In my opinion, the authors need to clarify, at least in this drug, the data they have for timing of approval and the contribution of the phase I pure pediatric trial to this approval. Would also be worth to check for the rest of the drugs, so that the methodology is consistent.

- Also important, is whether the studies they mention as reference for approval time after, where part of the PIP or the Written Request

- I do not clearly see the interest in the metric: Time from publication to regulatory approval. The marketing authorization expedited by the Competent Authorities (CA) relies on the Authorization Case File presented by the Sponsor that includes all administrative documentation, expert’s report, biological, chemical and pharmaceutical information, data from clinical trials, etc. This has not necessarily to include the publication, and therefore the approval may happen regardless the time the manuscript is published. Publication of the manuscript may also be conditioned by other external issues, falsely giving an external variability to the metric the authors are measuring which is independent from the development process. As an example, the phase III trial that leaded to the approval for everolimus in SEGA patients from 0 years of age was published in 2013 in NEJM, whereas the approval was one year earlier. The authors may consider, in turn, the first time the results were presented in a meeting for instance, as they are usually posted time ahead the final publication is issued, or the date the drug was approved for adults or the time when the first trial was initiated in adults, etc.

- This same concern about the interest of this metric makes me wonder whether the authors could re-consider the 5 year gap between publication and analysis of approval in the methodology of the study, which in my opinion, is not a methodological strong reason to exclude other drugs out of this 5 year period. By enlarging this time-period analysis the authors may be able to have a real median time that takes for a drug to be approved in children.

- In this table I am also missing asparaginase Erwinia chrysanthemi, that gained FDA approval first in 2011 and then in 2014, based on phase I/PK pediatric data (Salzer, Blood 2010). In the European side, there are also a few drugs that gained approval or changed label (e.g. mercaptopurine, mifamurtide) that may not fulfill the criteria for being analyzed as per the methodology of this study, but sure need to be commented to inform about the % of drugs approved in children compared to those identified in this study. For this, the authors may want to check this document: https://ec.europa.eu/health/sites/health/files/files/paediatrics/2016_pc_report_2017/ema_10_year_report_for_consultation.pdf. Table 27).

-Please consider all these issues in the limitations of the study.

RESULTS:

-LINE 184: Translation success. Again in this part the authors want to take into consideration the data from Neel D, Timing of first-in-child trials of FDA-approved oncology drugs. Eur J Cancer 2019. In this manuscript this same issue about hem Vs solid tumors is raised (or maybe to consider for the discussion).

-If the authors are doing any sort of statistical analysis between variables (as the p value they present in table 3), it needs to be depicted in the M&M methods. Besides this, the numbers are small, and split into many categories when they try to find a correlation, then it is very plausible that this is not significant anymore. This needs to be raised in the discussion as a limitation of the study.

-A complete reference of the translational success rate of the 139 studies is desirable to be added to the manuscript (supplemental table) so that the reader can check this information and see how many went into a phase II and how many did not.

-LINE 304: The sentence “Our review analyzed trials published between January 1st 2004” seems to be incomplete.

Reviewer #4: The manuscript “Clinical development success rates and social value of pediatric phase 1 trials in oncology: a meta-analysis” written by Mateusz Wasylewski and colleagues describes the clinical development success and social value of phase 1 trials in oncology, based on 1) translational success rate (registration of the tested drugs by FDA/EMA), 2) transition rate (number of trials that continued to phase 2/3) and 3) citation patterns.

Between January 2004 and December 2013, 139 pediatric phase I trials were identified. Of them 7/139 (5%) had their drugs FDA/EMA registered, 62/139 (45%) continued with phase 2/3 and over 90% were cited in the following years from publication. This study is a continuation of the previous systematic review with meta-analysis published by the same authors in 2018 based on the same studies but focused on objective reponse rate (ORR) and adverse events (AE).

Comments:

- In the last 10 years, several trials have been directly developed as phase 1/2 (dose escalation, expansion cohort, efficacy cohorts). How has this been addressed in the current trials reported in this review? It is important to clarify it as this may impact on the “transition rate” outcome.

- The current selection of studies until Dec 2013 limits the interest of the study has many of the main positive trials with targeted therapies in children with cancer have been published or shared in cancer meetings in the last 3-4 years (Ceritinib ASCO 2015, Sonidegib Neuro-Oncol 2017, Dabrafenib Clin Cancer Res 2019, Larotrectinib/Entrectinib ASCO 2019).

- Primary outcome is the translational success rate. However; this can be quite criticize (as presented in the Discussion section) as not always proven efficacy of a specific therapy is correlated to FDA/EMA approval (crizotinib is a good example). Probably the best outcome is ORR/AE (already published in 2018 by the same authors). Another outcome worth analyzing would be progression-free survival.

The manuscript is well written with a good methodology and has not been published elsewhere and conclusion are supported by the presented data. However the added value to the pediatric oncology community is strong but limited (compared to the already published review in 2018) and I would suggest to the authors to refer it as a “letter to the editor”.

6. PLOS authors have the option to publish the peer review history of their article (what does this mean?). If published, this will include your full peer review and any attached files.

Reviewer #1: Yes: Benjamin Kasenda

Reviewer #2: Yes: Tatiane B. Ribeiro

Reviewer #3: No

Reviewer #4: No

---

## [Author Response · Author response to Decision Letter 0]

24 Apr 2020

Dear Editor, Dear Reviewers,

Thank for your all useful comments on our manuscript. We have addressed them in the current version and we believe the manuscript has been improved. We clarified certain points and considered further limitations of our study pointed out by the reviewers. All changes made to the original version are highlighted in a marked up copy of our manuscript.

Please find our responses below.

Yours sincerely,

Marcin Waligóra

Reviewer #1: Waligora and colleagues submitted an investigation of the social and scientific value of pediatric phase I studies based on a previous systematic review of the group. The manuscript is well written and clear. The provided information is of value to the scientific community. I only have some minor comments that hopefully help to improve the manuscript

1. In your section statistical analysis, you mention to use techniques of meta-analysis to investigate the impact of previous EMA or FDA approval, but I cannot find that in your results section. Please clarify, because this would also impact on your manuscript title. Where is the meta-analysis?

Response: Our original manuscript was subtitled “A Meta-Analysis.” On reflection and prompted by these comments, we removed that subtitle, as the study was building on a previous meta-analysis and the techniques for the primary analysis were not meta-analytic in the strict sense. As for the statistical section: we had performed both a meta-analysis and a Poisson regression as an exploratory analysis. This was done to model the effects of whether a drug was previously approved by EMA and/or FDA on objective response (meta analysis) as well as adverse events (Poisson regression). However, these analyses were not directly related to the main objective of the study and we decided not to include them in our manuscript. We had mistakenly left some traces of this analysis; it has now been removed. In case the reviewer finds it relevant we have pasted below the original Poisson regression and meta-analysis. We will deposit these analyses in the Open Science Framework.

 Objective response rate

(95% CI) p value for group comparison Average rate of AEs per person (95% CI) p value for group comparison

not previously approved 0.0306 (0.0214-0.0397) 0.003 0.77 (0.72-0.82) <0.001

previously approved by both agencies 0.1424 (0.0763-0.2085) 0.99 (0.93-1.06) 

previously approved only by FDA 0.1504 (0.1043-0.1965) 2.06 (1.98-2.15) 

After adjustment for type of therapy

not previously approved 0.078 (0.030-0.127) 0.001 0.97 (0.90-1.06) <0.001

previously approved by both agencies 0.147 (0.092-0.201) 1.27 (1.05-1.53) 

previously approved only by FDA 0.132 (0.091-0.172) 2.16 (2.06-2.27) 

After adjustment for type of tumor

not previously approved 0.02 (0.0-0.933) 0.01 0.50 0.43-0.57 <0.001

previously approved by both agencies 0.061 (0.0-0.131) 0.69 (0.62-0.80) 

previously approved only by FDA 0.087 (0.013-0.131) 1.40 (1.12-1.57) 

After adjustment for number of drugs

not previously approved 0.031 (0.0-0.0657) 0.001 0.68 (0.63-0.72) 0.001

previously approved by both agencies 0.111 (0.0687-0.153) 0.92 (0.86-0.98) 

previously approved only by FDA 0.582 (0.01-0.107) 1.40 (1.31-1.50) 

The overall response was calculated using meta analysis

The overall adverse events were calculated using Poisson regression

2. Have you considered conducting a time to event analysis? E.g. time from publication of the phase I study to citation? By this you could additionally and easily visualize your findings.

Response: Thank you for this suggestion. We did not extract data on an exact time of every citation, thus we are not able to conduct time to event analysis. We considered time to event from publication of Phase 1 to the FDA approval, but the number of approvals was too low to support a meaningful analysis.

3. I suggest to provide more information on the type of drugs; not only for those that have been approved by FDA/EMA later on, but for all that have been tested in your included phase I trials.

Response: We included a table with information about drugs in the supplementary materials (S1 and S2 tables; https://osf.io/36prm/, https://osf.io/tzh8q/) We call out this additional information in the results section, on pages 9 and 10.

4. “Being previously approved for adults by FDA or EMA had no significant effect on progression to later phases of trials (p=.657 and p=.28 for ClinicalTrials.gov and EU Clinical 207 Trials Register respectively).” Please also report the frequencies with percentages, not only the p-values. You do that in the tables, but would be great to also see that in the text.

Response: We now report data as both percentages and p values on page 11 in the results section.

Reviewer #2: This paper included a sample of 139 pediatric phase 1 trials in oncology, previously found in a systematic review that retrieved studies from 2004 to 2013 (Waligora et al., 2018), in order to assess:

i. Approval rate (FDA or EMA) - named by the author as translational success,

ii. Rate of advance to further phases (phase 2, 3 or 4) - named by the author as transition success and,

iii. Citation patterns.

The authors found that 5% of pediatric phase 1 trials in oncology turned to approval in the FDA or EMA, despite a large proportion which have been tested in phase 2, 3 or 4. The trials were widely cited in primary research suggesting a scientific and social impact.

This is a very interesting research topic because pediatric population is vulnerable and few trials have been carried out in this population. Lack of published data increase the off-label use and evidence-based oriented practices is a challenge. United States of America and European Union are the head of a movement to increase pediatric research with special fundings and programs.

The paper aim is very important and open a discussion to understand the barriers to pediatric drug development.

1. I suggest improvements in the text and the methodology in order to clarify the reader and increase transparency:

Title - In my opinion, this paper did a statistical analysis with the data available, not a meta-analysis. A meta-analysis is defined as “a statistical procedure that integrates the results of several independent studies considered to be “combinable” (Egger M, Smith GD, Phillips AN, 1997). I suggest the title review;

Response: Thank you for this comment. We agree, and the first referee flagged a similar issue – please see the response to Reviewers’ #1 point 1 above.

2. Abstract/Objective - It would be more informative for the reader the definition of ‘translation’ or changing the term ‘translation’ for “approval rate in FDA or EMA”;

Response: The term “translation” has been changed to “approval rate” in the abstract and throughout the main text.

3. Abstract/Methods - Clarify for the reader that the cohort of 139 studies was systematically searched in a previous study (Waligora et al., 2018), in my opinion, it’s important to mention in the Abstract only the methodology of the current paper.

Response: Thank you for this comment. We clarified this by stating: “The current analysis assessed all studies in that review published between January 1st 2004 and December 31st 2013 for predictors of social value. This time range allowed for at least five years of subsequent development activity”. 

4. It is unclear to the reader this paragraph “We included small sample size, non randomized, dose escalation pediatric cancer phase 1 studies of any malignancy, that assessed chemotherapy and/or targeted therapy and were published between January 1st 2004 and December 31st 2013...”. Is this the inclusion criteria from the previous study? How was it related to other databases searched, FDA and EMA medicine databases, ClinicalTrials.gov and EU Clinical Trials Register or Google Scholar?

If this is the inclusion criteria of the previous systematic review, it would be more clear to maintain it in the beginning of the methodology.

Small sample isn’t an inclusion criteria in the paper methods.

Response: We included Phase 1 studies in pediatric oncology defined as “small sample size, non randomized, dose escalation studies”. Moreover there were additional inclusion criteria for those studies, mentioned both in the protocol of our review as well as in the manuscript (“studies of any malignancy, assessing chemotherapy and/or targeted therapy and looked at risk and benefit. The current analysis assessed all studies in that review published between January 1st 2004 and December 31st 2013”). The aim of our review is to investigate how many of such defined studies were approved, advanced to further phases and cited. 

We defined these inclusion criteria in our previous review (where we tested risk and benefit of those kind of studies). In current review we are using these criteria again. 

We clarified this in a current version of the manuscript as follows: “We performed an analysis on a subset of eligible trials included in a previous meta-analysis. That study systematically searched EMBASE and PubMed for small sample size, non randomized, dose escalation pediatric cancer Phase 1 studies of any malignancy, assessing chemotherapy and/or targeted therapy and looked at risk and benefit. The current analysis assessed all studies in that review published between January 1st 2004 and December 31st 2013 for predictors of social value. This time range allowed for at least five years of subsequent development activity. Sources of data included FDA and EMA medicine databases (for approval), ClinicalTrials.gov and EU Clinical Trials Register (for transition) and Google Scholar (for citation)” (page 2, methods section of the abstract). We hope that the Reviewer finds this clearer.

Methods

5. I suggest that the authors clarify what is part of the previous study and what are the methods of the current one. Thus, I suggest that all the information regarding the systematic review, Waligora et al., 2018, remain in the session ‘Study selection and eligibility criteria’ (Pg4/Line73).

Moreover, a critical aspect of this study is the usage of 3 characteristics (‘Type of tumor’, ‘Number of drugs’ and ‘Drug/s generally approved by FDA or EMA before study publication’) in the results and for statistical analysis. However these topics were not mentioned in the methods. The authors should explain the rationale for this 3 characteristics inclusion and its impact in the aim and results of the study.

-Pg4/Line73 - It would be important to mention the databases searched in the previous systematic review (The authors mentioned it in the Abstract).

Response: We address each of the above suggestions in revisions.

6. -Pg4/Line76 - I suggest a Phase 1 definition as their prior paper “Phase I studies, defined as “small sample size, non-randomized, dose escalation studies that defined the recommended dose for subsequent study of a new drug in each schedule tested”” (Waligora et al., 2018)

Response: Thank you for this suggestion; we included this definition in our methods.

7. -Pg4/Line80 - It would be important to reference the pediatric age range, because it can diverge, e.g. the FDA consider pediatrics individuals under 18 years old and the American Academy of Pediatric define pediatrics individuals under 21 years old.

Response: We agree. However, reporting of age in studies was highly variable and did not allow us to calculate age range. Our included trials were considered as pediatric if all or most participants were below 21 years of age. Choosing 21 years as an age threshold enabled us to encompass both FDA and EMA (as well as AAP) definitions of pediatric participants. We clarified this in the manuscript now (“choosing 21 years as an age threshold enabled us to encompass both EMA and FDA definitions of pediatric participants”, see page 5, the study selection and eligibility criteria section)

8. - Pg5/Line96 and Pg5/Line98-104 - Information about previous study and criteria for selection, I suggest to move to ‘Study selection and eligibility criteria’ (Pg4/Line73). How many studies were assessed, 170 or 139? In my understanding all outcomes should use the same timeframe previously specified from 2004 to 2013. The PRISMA flow in the Supplemment material made it clear however, in the text it would be important to fill this gap to the reader understanding.

Response: We edited the manuscript as suggested and included PRISMA in the main body of the manuscript. 

9. - Pg5/Line96-98 “After determining the registration….(Table 1) - Table 1 and this paragraph are results. Describe the methods which information the authors will extract from the papers.

Response: We clarified this in a new version of the manuscript in the Methods section.

10. -Pg9/153 - I suggest an independent topic about ‘Data extraction’

Response: Good idea. We edited manuscript and added a section to the Methods section titled ‘Data Extraction’.

11. -Pg9/158-9 - The authors mentioned data extraction related to: “study design, funding, intervention, outcomes and the timing of pediatric testing relative to adult testing as well as data on response and toxicity”. However only funding was reported as result. Please check data extraction.

Response: We reported most of these data merely as a study characteristic, not as a result. We deleted the sentence mentioned by the Reviewer and kept these data only at the Table 1.

12. In the results, it was collected data and performed statistical analysis for ‘Type of tumor’, ‘Number of drugs’ and ‘Drug/s generally approved by FDA or EMA before study publication’ for each outcome (i. Approval rate, ii. Posterior phases rate and iii.Citation pattern), despite it, none of these information were mentioned in the method.

Response: We included this information in the method section now. The data extraction paragraph now features the following: “In our previous systematic review of risk and benefit of Phase 1 trials in pediatric oncology we created a database of information and outcomes extracted from each study. We retrieved from this database data on study design, funding, intervention, outcomes and the timing of pediatric testing relative to adult testing for every trial included in the current analysis. For the outcomes of approval rate, transition rate and citation patterns we created and piloted a new extraction form”.

13. In the PRISMA figure at the Supplemental material and reading Waligora et al., 2018, it’s clear that 139 studies is from 2004 to 2013 and 170 the number retrieved in the previous study from 2004 to 2013.

We included PRISMA in the main body of the manuscript now to make it clear.

Results

14. -Pg10/173 - Clarify that ‘One hundred and thirty-nine Phase 1 pediatric oncology trials’ was retrieved from the past study.

Response: We clarified this in a new version.

15. - Pg12/209 - In Table 4, I suggest to merge information about Clinicaltrials.gov and EU Clinical Trials Registered. I understood that the analysis aims to access transition to further phases and not compare the protocol-registry databases. Moreover, it’s mandatory the previous registration in the database linked to the trial location, so ClinicalTrials.gov would be mandatory to studies carried out in the USA and EU CT in Europe. Thus, it would be more useful to show the data in a single column, Transition success (advance to further phases in Clinicaltrials.gov OR EU Clinical Trials Registered.)

Response: We prepared a new table, where we show transition outcomes merged together for all databases. We acknowledge the validity of Reviewer’s point

Discussion

16. Pg12/227 - The authors might describe Phase 1 trials in details, ‘pediatric Phase 1 trials in oncology’.

Response: We changed this as suggested: “The overall percentage of Phase 1 pediatric oncology trials included in our study that resulted in a regulatory approval was approximately 5%” (page 13, beginning of the discussion section).

17. Pg13/234 - It would be interesting if the author related this increase in single drugs approval with increase in target treatment in oncology that are in the great majority, a single drug. Multiple drugs approval are associated with conventional chemotherapy cytotoxic drugs, which have very few approvals in the past years.

Response: Thank you for this suggestion. We mention this as a possibility in our manuscript, though we also note that there are many combo trials aimed at synthetic lethality (using two targeted drugs) and the volume of immunotherapy-targeted drug trials is enormous in the adult literature.

18. Pg13/242 - This finding isn’t clear in the results.

Response: The finding mentioned by the Reviewer (“We however did not find any significant relationship between previous approval for adult population and probability of subsequent drug approval for children in our analysis”) was based on the p value for differences within categories provided by Fisher's test reported in Table 2 detailing the rate of approval for pediatric population by FDA or/and EMA. This is now called out in the results text. 

19. The discussion is interesting and I additionally suggest that the authors include more discussion about the link between their results and the opportunities and challenges for pediatric research development.

Response: Thank you for the suggestion. We included parts discussing current trends in the drug development (targeted therapies and cytotoxic drugs, single and multiple agents). We believe that discussion covers some challenges and opportunities for pediatric drug development however we would be happy to add any additional particular points suggested by the reviewer.

Reviewer #3: The authors present the results of a meta-analysis of Phase I pediatric oncology clinical trials. This current study is based on a previous systematic review of risk and benefit of pediatric Phase I clinical trials. From that study, the authors have selected 139 studies for further analysis to assess surrogates of social value of Phase I trials in pediatric cancer patients.

The idea is interesting and the effort from the authors is laudable.

I have some comments for the authors.

INTRODUCTION:

1. LINE 46: When the authors state that there is not for oncology agents targeting pediatric patients, they may want to consider adding this reference: Neel D, Timing of first-in-child trials of FDA-approved oncology drugs. Eur J Cancer 2019.

Response: Thank you for this useful reference, which we had missed. Now referenced. 

METHODS:

2. Beyond the 139 selected studies I am missing a few Phase I trials that may fulfill the eligibility criteria but are not included within this short list (Table S3). The following are only some examples:

- Caruso DA, Orme LM, Neale AM, et al. Results of a phase 1 study utilizing monocyte-derived dendritic cells pulsed with tumor RNA in children and young adults with brain cancer. Neuro Oncol. 2004;6:236-246.

- Kramer K, Humm JL, Souweidane MM, et al. Phase I study of targeted radioimmunotherapy for leptomeningeal cancers using intra-Ommaya 131-I-3F8. J Clin Oncol. 2007;25:5465-5470.

- Gilman AL, Jacobsen C, Bunin N, et al. Phase I study of tandem high-dose chemotherapy with autologous peripheral blood stem cell rescue for children with recurrent brain tumors: a Pediatric Blood and MarrowTransplant Consortium study. Pediatr Blood Cancer. 2011;57:506-513.

Response: Thank you for this suggestion. Our paper was based on a cohort of trials predefined in our former review (PROSPERO CRD42015015961). Trials suggested by the reviewer above did not fulfill inclusion criteria in that study. For example, we excluded studies where surgery and radiotherapy were the only treatment (radioimmunotherapy studied in Kramer 2007) or were used together with chemotherapy/targeted therapy. We also excluded supportive care without anticancer agents and other types of drugs and treatments i.e. antiviral agents or non-specific immunotherapy. That study also excluded cell therapies (Caruso 2004, Gilman 2011), because the nomenclature of cell therapies makes it difficult to trace lineages of clinical trial phases for the same product. We now call attention to this in the methods and note this as a limitation: “Second, our analyses do not include all types of trials: we excluded antiviral agents, non-specific immunotherapies, cell therapies and other agents when tested together with surgery or radiotherapy”. We hope this is clearer now.

3. In Table 1, the authors present a list of drugs that are approved for pediatric indications. In this table, the authors need to:

-In the column type of malignancy depict only the disease for which the drug is approved in children. For instance: Tablets for oral suspension of everolimus was granted by the FDA in 2012 for the treatment of pediatric and adult patients with tuberous sclerosis complex (TSC) who have subependymal giant cell astrocytoma (SEGA) that requires therapeutic intervention but cannot be curatively resected, but not the rest of the diseases that the authors describe in this column.

Response: The malignancies featured in Table 1 were the ones tested in the referenced trial, not the specific ones for which the drug is or was approved in children. The table has been changed to clarify that Please note that it has now been removed from the manuscript. It is available on the OSF platform, as supporting information S1 Table (https://osf.io/36prm/).

4. -For each drug, to add a reference in the table supporting the information about the specific approval indication

Response: The relevant references have been added to S1 Table in the supporting information (https://osf.io/36prm/). Now also called out in the text on page 10 in the results section.

In this same table I have some comments:

5. - For the imatinib indication in pediatric brain tumors, can kindly the authors confirm the source of this information? I have not been able to find that imatinib is approved for such a number of indications as stated in this table. For double checking I have used this source:https://www.fda.gov/drugs/resources-information-approved-drugs/hematologyoncology-cancer-approvals-safety-notifications

Response: The malignancies featured in the Table 1 were the ones tested in the referenced trial, not the ones for which the drug is approved. The table has been changed to clarify that.

To check the first pediatric approval and its timing, we searched the FDA’s Drugs@FDA database for highlights of prescribing information documents, as well as EMA’s summaries of product characteristics available on the EMA website. According to the available Imatinib prescribing information, it was indeed approved for pediatric patients with Ph+ CML in chronic phase who are newly diagnosed or whose disease has recurred after stem cell transplant or who are resistant to interferon-therapy. Approved indications have been broadened since then, but for the purpose of our study it was sufficient to note that the first clear pediatric indication was present in the 2007 prescribing information.

6. - For the everolimus indication, the phase I from Fouladi did not include patients with SEGA, which is the final indication for this drug. This study, nevertheless, could contribute to the approval in 2012 of the everolimus tables for oral formulation, but in reality, this drug was already approved in 2010 for patients 3 years and older based on the pediatric and adult phase I/II trial (Krueger et al, NEJM 2010, NCT00411619). Furthermore, the approval in 2012 that the authors mention, is not based on the Phase I trial from Fouladi but rather in the Phase III randomized trial including children and adults (Franz, Lancet 2013, NCT00789828). In my opinion, the authors need to clarify, at least in this drug, the data they have for timing of approval and the contribution of the phase I pure pediatric trial to this approval. Would also be worth to check for the rest of the drugs, so that the methodology is consistent.

Response: Thank you very much for these insights. The relevant references supporting the information about specific approval indication have been added to S1 Table.

Unfortunately the FDA and EMA documents consulted during the extraction process often do not contain definite references to trials or data on which the approval in question was based. Therefore we did not have an objective way of reliably linking trials assessed in our study to the corresponding approvals. Instead, we simply determined whether a given regimen was FDA approved for a given indication- and made no judgments about whether the specific Phase 1 trial had been considered by regulatory authorities. We note this may have led to a slight overestimate of impact, since it is possible some FDA approvals were motivated by data other than the phase 1 trial. Now stated more clearly in the methods (“Drugs were classified as approved for pediatric use if there was a clear reference to pediatric population in “indications” or “dosing” sections of regulatory documents. If a Phase 1 study tested more than one drug, approval status for all drugs was determined. The trial regimen was then classified as “successful” if the whole combination (all of the investigated drugs) were approved by FDA and/or EMA after study publication.”).

Regarding the specific trials and approvals above: the 2010 FDA “Highlights of prescribing information” document did not contain a specific pediatric indication. Only the 2012 and subsequent labels have a pediatric indication. That is why we classified everolimus’ pediatric approval as occurring in 2012. The Kruger 2010 study was not included into a cohort of our studies from previous systematic review, because it did not report Phase 1 trials results separately (and this was an inclusion criterion).

Again, the aim of our current analysis was to take a cohort of pediatric Phase 1 oncology trials and work forwards to probe for indicators of broader social value. Our aim was not to work retrospectively - taking approvals themselves and working backwards to the studies (mixed, pediatric, etc.) that led to them.

7. - Also important, is whether the studies they mention as reference for approval time after, where part of the PIP or the Written Request

Response: Unfortunately, this was outside the scope of our study. It would be valuable to look at this in the future and see if it contributed to a faster approval. The time to approval metric was not analyzed in our study (please see also response to the next comment).

8. - I do not clearly see the interest in the metric: Time from publication to regulatory approval. The marketing authorization expedited by the Competent Authorities (CA) relies on the Authorization Case File presented by the Sponsor that includes all administrative documentation, expert’s report, biological, chemical and pharmaceutical information, data from clinical trials, etc. This has not necessarily to include the publication, and therefore the approval may happen regardless the time the manuscript is published. Publication of the manuscript may also be conditioned by other external issues, falsely giving an external variability to the metric the authors are measuring which is independent from the development process. As an example, the phase III trial that leaded to the approval for everolimus in SEGA patients from 0 years of age was published in 2013 in NEJM, whereas the approval was one year earlier. The authors may consider, in turn, the first time the results were presented in a meeting for instance, as they are usually posted time ahead the final publication is issued, or the date the drug was approved for adults or the time when the first trial was initiated in adults, etc.

Response: We agree. The only place where this metric was presented was Table 1. We removed it. We also add a limitation in the discussion about the variability of publication timing (“It is possible, however, that some regulatory approvals or phase transitions might have occurred after a Phase 1 trial was complete but before it was published”). Ideally we would have used 5 years since study close, not publication, to establish our cohort. However, this datum was not presented in many Phase 1 trial publications, rendering such a preferred approach impossible.

We appreciate the point and agree that any number of variables affect timing of publication (publication can in principle come even after approval, if ever!). First, our study did not involve a time to event analysis. Rather, we defined our cohort based on having 5 years of follow up from publication. As described in our methods and discussion sections, this time frame was based on a retrospective analysis of FDA pediatric approvals, and was sufficient enough to capture all FDA approvals that occurred. However, we acknowledge that there could be cases where FDA approval came after a study was complete but before its publication. Now noted in the discussion (“some regulatory approvals or Phase transitions might have occurred after a Phase 1 trial was complete but before it was published. As such, our estimates might have undercounted some approvals and transitions ”)

Ideally, our study would have worked from timing of trial close of enrollment to regulatory approvals, etc. However, trials did not always state when they ended enrollment, making such an analysis impossible. 

Moreover, the time from publication to regulatory approval was not used as a metric in our outcome analysis. All approvals in our previous sample of studies (170 trials assessed in previous systematic review) occurred within a maximum of 5 years since study publication (i.e. there were no approvals after 6, 7 or 8 years for any study, at least till the beginning of 2018, when the extraction was finalized).

We used this fact as a justification for choosing a 5 year follow up threshold for the two other outcomes (transition and citation). This allowed us to determine the impact of reviewed studies on social value in the same time frame for all 3 outcomes (approval, transition and citation).

Unfortunately, settling on this follow up threshold meant restricting our sample to 139 studies, for which said amount of follow up was available. The “time lag between publishing and changing label” column was put in the table detailing trials that led to regulatory approval to help clarify that, but time to approval was not used as a direct metric in our analysis.

9. - This same concern about the interest of this metric makes me wonder whether the authors could re-consider the 5 year gap between publication and analysis of approval in the methodology of the study, which in my opinion, is not a methodological strong reason to exclude other drugs out of this 5 year period. By enlarging this time-period analysis the authors may be able to have a real median time that takes for a drug to be approved in children.

Response: We appreciate the suggestion. However as described above, we did not limit the follow up of approval to 5 years – the follow up was only limited for transition and citation endpoints. This was done so that every study had the same chance to have impact on social value, as assessed by these two predictors. Had we used a longer follow up time, more of our data would need to be censored. Had we used a shorter follow up time, we would worry we did not give treatments a proper chance to get FDA/EMA approval or to graduate. We address this in our discussion.

10. - In this table I am also missing asparaginase Erwinia chrysanthemi, that gained FDA approval first in 2011 and then in 2014, based on phase I/PK pediatric data (Salzer, Blood 2010). In the European side, there are also a few drugs that gained approval or changed label (e.g. mercaptopurine, mifamurtide) that may not fulfill the criteria for being analyzed as per the methodology of this study, but sure need to be commented to inform about the % of drugs approved in children compared to those identified in this study. For this, the authors may want to check this document: https://ec.europa.eu/health/sites/health/files/files/paediatrics/2016_pc_report_2017/ema_10_year_report_for_consultation.pdf. Table 27). -Please consider all these issues in the limitations of the study.

Response: The asparaginase mentioned by the Reviewer was studied in several trials included in our review. Its approval status was checked, but the studies in question (Nelken 2012, Messinger 2010 and Aplenc 2008) were not classified as leading to approval. As per our criteria, this was due to the fact that not the whole combination used in the trial was approved for pediatric use after study publication. This does not mean that asparaginase itself was not approved, only that we could not classify the Phase 1 trial as leading to approval.

Phase 1 trials with mercaptopurine and mifamurtide did not fulfill the criteria for inclusion into our systematic review (the previous study), the possible reasons are, for example, that they were not published in 2004-2014), that they were reported together with surgery or radiotherapy or other reasons defined in our exclusion criteria. Unfortunately we are not able to provide definite reasons on that stage of our study (we would need to re-examine a portion of 6409 studies excluded after our search). Instead, after the reviewer’s inquiry, we performed an exploratory search in Google Scholar and we did not find Phase 1 studies with mercaptopurine or mifamurtide performed between 2004-2014. And this is the most the most probable cause of exclusion. Because the trials with those drugs were not included in the previous sample of Phase 1 trials, they were not analyzed as per the methodology of the current study.

The scope of our study was to assess chemotherapy and targeted agents, used as monotherapy or in combination with each other. The purpose of our study was to assess different predictors of social value of included studies, not do detail the approval process for each pediatric oncology interventions. We hope that the reviewer agrees and accepts this. 

RESULTS:

11. -LINE 184: Translation success. Again in this part the authors want to take into consideration the data from Neel D, Timing of first-in-child trials of FDA-approved oncology drugs. Eur J Cancer 2019. In this manuscript this same issue about hem Vs solid tumors is raised (or maybe to consider for the discussion).

Response: We included this reference now.

12. -If the authors are doing any sort of statistical analysis between variables (as the p value they present in table 3), it needs to be depicted in the M&M methods. Besides this, the numbers are small, and split into many categories when they try to find a correlation, then it is very plausible that this is not significant anymore. This needs to be raised in the discussion as a limitation of the study.

Response: We agree and state that we did not adjust for multiple comparisons (see methods). Fisher’s exact test we used is designed for small samples. 

13. -A complete reference of the translational success rate of the 139 studies is desirable to be added to the manuscript (supplemental table) so that the reader can check this information and see how many went into a phase II and how many did not.

Response: We prepared a new supplementary appendix containing the information requested by the reviewer (S2 Table, https://osf.io/tzh8q/); now called out on page 9 in the results section. 

14. -LINE 304: The sentence “Our review analyzed trials published between January 1st 2004” seems to be incomplete.

Response: Thank you. We corrected the sentence now. 

Reviewer #4: The manuscript “Clinical development success rates and social value of pediatric phase 1 trials in oncology: a meta-analysis” written by Mateusz Wasylewski and colleagues describes the clinical development success and social value of phase 1 trials in oncology, based on 1) translational success rate (registration of the tested drugs by FDA/EMA), 2) transition rate (number of trials that continued to phase 2/3) and 3) citation patterns.

Between January 2004 and December 2013, 139 pediatric phase I trials were identified. Of them 7/139 (5%) had their drugs FDA/EMA registered, 62/139 (45%) continued with phase 2/3 and over 90% were cited in the following years from publication. This study is a continuation of the previous systematic review with meta-analysis published by the same authors in 2018 based on the same studies but focused on objective reponse rate (ORR) and adverse events (AE).

Comments:

1. - In the last 10 years, several trials have been directly developed as phase 1/2 (dose escalation, expansion cohort, efficacy cohorts). How has this been addressed in the current trials reported in this review? It is important to clarify it as this may impact on the “transition rate” outcome.

Response: Thank you for this suggestion. We did not capture seamless Phase 1 trials and the like in this study - though we also note that most trials in our sample precede the emergence of these methods. Subsequent work by a member of our team (JK) shows such methods were not used widely in pediatric cancer in the time period of the trials reflected in the present paper. We now describe this as a limitation of our study: "The fact that some recent cancer drugs have been directly developed using Phase 1/2, seamless trial approaches may be another limitation of our approach to estimating transition success [Hutchinson 2020], though we are not aware of any pediatric drug approvals in this period that resulted from a seamless approach” 

2. - The current selection of studies until Dec 2013 limits the interest of the study has many of the main positive trials with targeted therapies in children with cancer have been published or shared in cancer meetings in the last 3-4 years (Ceritinib ASCO 2015, Sonidegib Neuro-Oncol 2017, Dabrafenib Clin Cancer Res 2019, Larotrectinib/Entrectinib ASCO 2019).

Response: We agree: cohort studies like ours are a challenge. But there is simply no alternative but to pick a cohort of trials that has a “fair” chance of advancing to a licensure in a reasonable time frame. To maintain integrity of analysis, our study must adhere to all methods (including dates of publication) defined in the protocol (PROSPERO 2018 CRD42018106213). A next meta-research would include more recent studies and possibly identify recent progress. Our view is that our data are better than no data (or reliance on anecdotal reports of cases like ceritinib). We also note that a) prior studies of drug development do not show dramatic changes in success rates or response rates over the last two decades, contrary to popular perception (see our previous meta-analysis, for example, and b) almost every study widely cited about drug development published in JAMA and the like (e.g. success rates, costs of developing new drugs, etc.) rely on similar cohort approaches. This point is acknowledge in the limitations section: “Our analysis has several limitations. First, our group of 139 trials involved 5 years of follow up from publication of the Phase 1 trial. On the one hand, changes in transition or approval rates occurring in the last five years will not be reflected in our estimates. On the other hand, any phase transition or influence on subsequent publications occurring outside that time frame would not be reflected in our analysis”.

3. - Primary outcome is the translational success rate. However; this can be quite criticize (as presented in the Discussion section) as not always proven efficacy of a specific therapy is correlated to FDA/EMA approval (crizotinib is a good example). Probably the best outcome is ORR/AE (already published in 2018 by the same authors). Another outcome worth analyzing would be progression-free survival.

Response: Thank you for this suggestion. We are not sure we understand what the referee is saying re: crizotinib. Perhaps he/she could clarify? Achievement of FDA/EMA approval, by any measure, is an important milestone in a drug’s development. We nevertheless agree it is imperfect: many useful treatments are used off label on the basis of response rates from clinical trials, and never get FDA approval, while many almost useless drugs nevertheless receive FDA approval. Readers can consult our previously published meta-analysis to find examples of Phase 1 pediatric trials that had very high response rates- here the goal is to track different measures of research success.

4. The manuscript is well written with a good methodology and has not been published elsewhere and conclusion are supported by the presented data. However the added value to the pediatric oncology community is strong but limited (compared to the already published review in 2018) and I would suggest to the authors to refer it as a “letter to the editor”.

Response: Thanks. There is a lot of data and analysis in our paper- a letter to the editor format would not give us sufficient space to present this detail.

---

## [Decision Letter · Decision Letter 1]

2 Jun 2020

PONE-D-19-32233R1

Clinical Development Success Rates and Social Value of Pediatric Phase 1 Trials in Oncology

PLOS ONE

Dear Dr. Waligora,

Thank you for submitting your manuscript to PLOS ONE. After careful consideration, we feel that it has merit but does not fully meet PLOS ONE’s publication criteria as it currently stands. Therefore, we invite you to submit a revised version of the manuscript that addresses the points raised during the review process.

Only a handful of very minor comments still remain to be addressed.

Regarding the reviewer's suggestion, the Supplementary Tables are already cited in-text, so it's ok. I don't think however you need to cite at instance of Supplementary tables the OSF link--the citation at the end of the manuscript about the supplemental material sufizes.

We look forward to receiving your revised manuscript.

Kind regards,

Spyridon N. Papageorgiou, DDS, Dr Med Dent

Academic Editor

PLOS ONE

Reviewers' comments:

Reviewer's Responses to Questions

**Comments to the Author**

1. If the authors have adequately addressed your comments raised in a previous round of review and you feel that this manuscript is now acceptable for publication, you may indicate that here to bypass the “Comments to the Author” section, enter your conflict of interest statement in the “Confidential to Editor” section, and submit your "Accept" recommendation.

Reviewer #2: All comments have been addressed

Reviewer #3: All comments have been addressed

2. Is the manuscript technically sound, and do the data support the conclusions?

Reviewer #2: Yes

Reviewer #3: Yes

3. Has the statistical analysis been performed appropriately and rigorously? 

Reviewer #2: Yes

Reviewer #3: Yes

4. Have the authors made all data underlying the findings in their manuscript fully available?

Reviewer #2: Yes

Reviewer #3: Yes

5. Is the manuscript presented in an intelligible fashion and written in standard English?

Reviewer #2: Yes

Reviewer #3: Yes

6. Review Comments to the Author

Reviewer #2: Dear all,

The revised manuscript included several alterations and improvements in the methodology and results presentation. The authors addressed many different aspects from the reviewer's comments and added valuable information!

My comments in the previous manuscript were fully responded and I am happy with the alterations.

I have only minor corrections before the publication:

1.pg 5 line91 - In my opinion, this is restricted to the Results session (it is not Methods).

"... We found that 6 drugs tested in 7 trials were approved by at least one agency for pediatric use in oncologic conditions (Supporting information S1 Table, https://osf.io/36prm/)"

2.pg 10-11 (Table2) - Please, revise all the values and percentages in Table 2 ("Total", "Number of drugs", "Drug/s generally approved by FDA or EMA before study publication". I calculated and the total was 135, instead of 139.

3. Great job with all supplementary content! In my opinion it might me included in PLOSONE platform as Supplementary material (not only at OSF). Thus, I suggest citation for Supplementary material, not OSF.

Kind regards

Tatiane

Reviewer #3: I would like to thank the authors for addressing my comments in their rebuttal letter and in the new version of the manuscript. I am happy with the new manuscript version.

7. PLOS authors have the option to publish the peer review history of their article (what does this mean?). If published, this will include your full peer review and any attached files.

Reviewer #2: No

Reviewer #3: No

---

## [Author Response · Author response to Decision Letter 1]

3 Jun 2020

Dear Editor, Dear Reviewers,

Thank you for accepting our previous revisions. We responses for reviewer’s comments below.

Reviewer #2: Dear all,

The revised manuscript included several alterations and improvements in the methodology and results presentation. The authors addressed many different aspects from the reviewer's comments and added valuable information!

My comments in the previous manuscript were fully responded and I am happy with the alterations.

I have only minor corrections before the publication:

1.pg 5 line91 - In my opinion, this is restricted to the Results session (it is not Methods).

"... We found that 6 drugs tested in 7 trials were approved by at least one agency for pediatric use in oncologic conditions (Supporting information S1 Table, https://osf.io/36prm/)"

Response: Thank you for this comment. We move this part to the Results section now. 

2.pg 10-11 (Table2) - Please, revise all the values and percentages in Table 2 ("Total", "Number of drugs", "Drug/s generally approved by FDA or EMA before study publication". I calculated and the total was 135, instead of 139.

Response: There was a clarification under the table “Four trials were excluded from the analysis due to not being applicable or an unclear status”. We now added a star to make this information more readable for the future readers. 

3. Great job with all supplementary content! In my opinion it might me included in PLOSONE platform as Supplementary material (not only at OSF). Thus, I suggest citation for Supplementary material, not OSF.

Response: Thank you! And many thanks for important suggestions!

Kind regards,

Marcin Waligora

---

## [Editor Report · Decision Letter 2]

5 Jun 2020

Clinical Development Success Rates and Social Value of Pediatric Phase 1 Trials in Oncology

PONE-D-19-32233R2

Dear Dr. Waligora,

We’re pleased to inform you that your manuscript has been judged scientifically suitable for publication and will be formally accepted for publication once it meets all outstanding technical requirements.

Kind regards,

Spyridon N. Papageorgiou, DDS, Dr Med Dent

Academic Editor

PLOS ONE
---

## [Editor Report · Acceptance letter]

15 Jun 2020

PONE-D-19-32233R2 

Clinical Development Success Rates and Social Value of Pediatric Phase 1 Trials in Oncology 

Dear Dr. Waligora:

I'm pleased to inform you that your manuscript has been deemed suitable for publication in PLOS ONE. Congratulations! Your manuscript is now with our production department. 

Kind regards, 

on behalf of

Dr. Spyridon N. Papageorgiou 

Academic Editor

PLOS ONE